# Thiol-Based Redox Molecules: Potential Antidotes for Acrylamide Toxicity

**DOI:** 10.3390/antiox13121431

**Published:** 2024-11-21

**Authors:** Valeria Martin, Michael Trus, Daphne Atlas

**Affiliations:** Department of Biological Chemistry, Institute of Life Sciences, The Hebrew University of Jerusalem, Jerusalem 91904, Israel; valeria.martin@mail.huji.ac.il (V.M.); michael.trus@mail.huji.ac.il (M.T.)

**Keywords:** oxidative stress, thioredoxin reductase, apoptosis, toxicity, AD4/NACA

## Abstract

**Highlights:**

**Abstract:**

*Acrylamide (ACR)* is a low-molecular weight, non-aromatic reagent, widely used in industry, such as in the manufacture of paper, textiles, plastics, cosmetics, and dyes. ACR is formed during the cooking of starchy food and its toxicity results mainly by conferring oxidative stress by elevating reactive oxygen species (ROS). To identify potential antidotes for ACR toxicity, we evaluated the efficacy of several thiol-based molecules known for ROS-scavenging, disulfide-reducing properties, and inhibition of oxidative stress-induced activation of the mitogen-activated protein kinases (MAPKs): the extracellular-signal-regulated-kinases (ERK1/2), p38-mitogen-activated-protein-kinases (p38^MAPK^), and c-Jun-N-terminal-kinases (JNKs). We established a reproducible assay testing N-acetylcysteine (NAC), AD4/NACA, and the N-and C-blocked tri- and tetra-thioredoxin-mimetic (TXM) peptides, in PC12 cells. Our results demonstrate that these compounds exhibited high efficacy in suppressing ACR-induced MAPK activation, either prior to or subsequent to ACR exposure. The inhibition by single cysteine (Cys) residue, NAC and AD4/NACA (NAC-amide), 2 Cys peptides TXM-CB30, AcDCys-Gly-DCysNH_2_, TXM-CB20, AcCys-Gly-CysNH_2_, SuperDopa (SD, Ac-CysL-Levodopa-CysNH_2_, TXM-CB13, AcCys-Met-Lys-CysNH_2_, and a 3-Cys peptide, TXM-CB16, AcCys-γGlu-Cys-CysNH_2_ was dose-dependent and potency displayed a direct correlation with the number of Cys residues. Cellular proteolysis of SD, which consists of levodopa flanked by two Cys, may suppress the manifestation of Parkinson’s disease (PD)-like symptoms mediated by chronic ACR exposure not only through lowering oxidative stress but also by replenishing cellular levels of dopamine. Overall, these results could advance the clinical application of TXM peptides as potential treatments for acute and/or chronic exposure to ACR and show promise as antidotes for preventing ACR-triggered PD-like neurotoxic symptoms.

## 1. Introduction

*Acrylamide (ACR)* (CH2=CHC(O)NH2) is the amide form of acrylic acid. It is widely used in industry, in particular in the manufacturing of paper textiles, plastic, cosmetics, and dyes. ACR is naturally formed during the cooking of starchy food. As a type-2 alkene, ACR displays electrophilic behavior. It can bind nucleophiles like DNA or protein motifs, in particular cysteine residues, forming toxic adducts, which are responsible for genotoxicity, and neurotoxicity [1,2,3,4].

Although the discovery of ACR in foods during heating, especially foods rich in carbohydrates, has drawn much attention [5], ACR levels found in cooked food are significantly low and it is not clear what risk these amounts pose to humans [6]. Since high doses of ACR have been shown to cause cancer, ACR has been defined as a ‘probable human carcinogen’, by the International Agency of Research on Cancer (IARC). Animal studies indicate that ACR is well absorbed, widely distributed in tissues, and is either metabolized to the reactive epoxide, glycidamide, or to glutathione (GSH) conjugates [7,8].

The cytotoxicity of ACR is primarily attributed to the disruption of the cellular redox state by increasing reactive oxygen species (ROS) levels and reacting with free thiol groups in antioxidants such as GSH, thioredoxin, and glutaredoxin. A recent study has shown that by reacting with GSH, ACR increases cellular vulnerability to ROS [9]. The molecular mechanism of ACR neurotoxicity involves its metabolic conversion into glycidamide (GA) by cytochrome P450 2E1, after which both ACR and GA conjugate with GSH [10]. Both ACR and GA possess the ability to form covalent bonds with sulfhydryl groups, thereby resulting in the formation of adducts that inactivate proteins vital for redox control of the cell [11]. The dysfunctional adduct-inactivated proteins disrupt the cellular redox balance. This in turn causes mitochondrial dysfunction and subsequent ROS production, which results in cytotoxicity and apoptosis [12]. ACR-induced ROS production was found to disrupt neurotransmission and axonal transport, resulting in neuronal cell death [12,13,14,15].

In addition to generating oxidative stress, ACR is also responsible for the endoplasmic reticulum stress-mediated apoptotic pathway. Exposure of SH-SY5Y human neuroblastoma cells to ACR showed an increase in levels of phosphorylated eukaryotic translation initiation factor 2α (eIF2α), and activation of transcription factor 4 (ATF4) [16], and chronic exposure of male mice to ACR has been shown to cause elevated DNA damage in the germline and heritable induction of CYP2E1 in the testes [17].

In HepG2 cells, ACR has been shown to induce NLRP3 inflammasome activation via oxidative and ER stress, mediated via the MAPK pathway [18]. A more recent study indicated that chronic exposure to ACR significantly increases the levels of the inflammatory cytokine IL-1β, the protein levels of NLRP3, inflammasome-mediated neuroinflammation, α-synuclein accumulation in the substantia nigra, and caspase activation [13]. These changes result in dopaminergic neuron loss, neuroinflammation, and motor impairment in rats.

Furthermore, several examples of Ser/Thr kinase regulation through redox-active Cys residues exist, including apoptosis signal-regulating kinase1 (ASK1), MEKK1, maternal embryonic leucine zipper kinase (MELK), protein kinase A (PKA), protein kinase G (PKG), and MAPK, ERK1/2, JNK, and p38^MAPK^ [19,20,21,22,23,24,25,26,27].

The abundance in foods rich in carbohydrates, the widespread use of ACR in industry and the potential for industrial spills or misuse as an agent of opportunity in chemical terrorism [28], have prompted extensive investigation into potential antidotes for both its acute and chronic effects [29]. One approach to counteracting ACR toxicity caused by an addition reaction with the reactive sidechain thiolate of Cys residues involves the use of antioxidants such as catechins, crocin, lipoic acid, and N-acetylcysteine (NAC) [14,30,31,32], which when administered as pre-treatment can prevent ACR-induced oxidative stress [33]. However, their effectiveness is limited due to low membrane permeability and an inability to cross the blood–brain barrier (BBB). Oxidative stress mediated by ACR appears to activate Nrf2 and/or NF-κB, which subsequently activates MAPK signaling [15,33].

The main objective of this project is to evaluate the potential of AD4/NACA and thioredoxin mimetic (TXM) peptides, which are BBB permeable drugs and potent antioxidants in mitigating ACR-induced activation of MAPK signaling pathways. The TXM peptides and AD4/NACA target oxidative stress, apoptosis, and inflammation, and effectively inhibit auranofin (thioredoxin-reductase inhibitor)-induced MAPK activation [34,35,36,37,38,39,40]. They exhibit anti-inflammatory properties by lowering oxidative-stress induced activation of NF-κB signaling pathways [41], lipopolysaccharide (LPS)-induced cytokines in mice [42], and the expression of cardiac inflammatory markers following myocardial infarction in vivo [43]. We established a reproducible cellular assay to monitor ACR-induced activation of extracellular signal-regulated kinases (ERK1/2), p38 mitogen-activated protein kinases (p38^MAPK^), and c-Jun N-terminal kinases (JNK1/2). Our findings revealed that members of the TXM peptide family inhibit ACR-induced MAPK activation more effectively than the single Cys antioxidants NAC and AD4/NACA. Overall, TXM peptides including SuperDopa (SD), a TXM-like peptide containing levodopa, appear to hold promise as effective antidotes against ACR toxicity, while SD is a promising candidate for reversing the parkinsonian-like symptoms associated with chronic ACR exposure.

## 2. Experimental Procedures

Anti-p-ERK 1/2 (Cell Signaling #9101S), Anti-p-p38 MAPK (Cell Signaling #9211L), Anti-p SAPK/JNK (Cell Signaling #9255S), Anti-pCREB (Cell signaling #4276), Anti-β-Catenin (Cell Signaling 37447S), Anti-mouse IgG, HRP-linked (Cell signaling 7076S), Anti-rabbit IgG, HRP-linked (Cell Signaling 7074S).

### 2.1. Chemicals and Reagents

2-Propenamide (Acrylamide, ACR; Sigma);NAC (Sigma); TXM-peptides (>98% pure Novetide, Ltd., Haifa, Israel); DMEM and L-alanyl-L-glutamine (Biological Industries, Kibbutz Beit-Haemek, Israel).

### 2.2. Cell Culture

Rat pheochromocytoma (PC12) cells (Dr. Rick Chris (New York, NY, USA)) were cultured in Dulbecco’s modified Eagle medium (DMEM) medium supplemented with 10% fetal bovine serum (FBS), Penicillin (100 U/mL), Streptomycin (1 µg/mL), L-alanyl-L-glutamine (3 mM). Cells were plated at a density of 8.5 × 10^4^/cm^2^ and incubated at 37 °C with 5% CO_2_. PC12 cells were divided into new flasks every 2–3 days, when they reached 80% confluency, at a ratio of 1:3–1:4. Cell batches were typically replaced after 16–18 passages by thawing a fresh vial from the expanded stock.

### 2.3. Acrylamide Toxicity Treatment Assay

PC12 cells were seeded in 24-well plates, cultured in 1 mL of medium overnight and treated for 2 h with 5 mM ACR. After replacing the medium, the cells were treated with increasing concentrations of AD4/NACA, NAC, or TXM-peptides for 1 h. Then, the cells were harvested with lysis buffer (Tris-Cl pH 6.8 150 mM, 10% glycerol, 2% SDS, 0.7% β-mercaptoethanol) and loaded on SDS-PAGE, or stored at −20 °C.

### 2.4. Acrylamide Toxicity Prophylaxis Assay

PC12 were seeded in 24-well plates, cultured in 1 ml media overnight and pre-incubated for 30 min with AD4/NACA or TXM-CB3. After changing the medium, the cells were preincubated for 2 h with 5 mM ACR. Then, the cells were harvested with lysis buffer (Tris-Cl pH 6.8 150 mM, 10% glycerol, 2% SDS, 0.7% β-mercaptoethanol) and loaded on SDS-PAGE, or stored at −20 °C.

### 2.5. Cell Viability and Methylene Blue Assay

PC12 cells were seeded in 96-well plates for 24 h and then treated with 2.5 mM ACR for 2 h, or DMEM, washed and co-cultured with or without increasing concentrations of NAC, AD4/NACA, or TXM-CB3. Forty-eight hours later, the cells were fixed with 2.5% glutaraldehyde in a final concentration of 0.5% for 10 min. The cells were washed 3X with DDW and dried overnight. The fixed cells were stained with 100 µL of 1% methylene blue (1% filtered methylene blue dissolved in 0.1 M boric acid pH 8.5) washed for 1 h, and dried for 1 h at 50 °C, or overnight. The color was extracted with 200 µL of 0.1 M HCl for 1 h at 37 °C, and read at a 630 nM in Plate Reader Synergy H1.

### 2.6. Western Blot Analysis

Samples of cell lysates (20–60 µg protein) were loaded, and proteins were separated on 10–12% SDS-PAGE. The proteins were then transferred electrophoretically to nitrocellulose (Whatman, Maidstone, VT, USA). The blots were blocked by incubation for 1 h at RT in TBS-T (25 mM Tris–HCl pH 7.4, 0.9% NaCl and 0.02% Tween-20) with 4% Difco skim milk (BD, Franklin Lakes, NJ, USA), and incubated over-night at 4 °C with the primary antibody: pERK1/2 (Thr 202/Tyr204), rabbit mAb; p-SAPK/JNK (Thr183/Tyr185), mouse mAb; p-p38MAP kinase (Thr180/Tyr182), rabbit mAb. Antibodies were diluted 1:1000, if not otherwise stated. Purified β-Catenin, mouse mAb, (1:10,000). Proteins were detected with anti-mouse or anti-rabbit IgG-HRP linked antibody (1:10,000).

### 2.7. Data Analysis and Statistics

All the data are presented as mean ± SEM from two independent experiments carried out in duplicates, normalized with β-Catenin. The densitometry was determined using ImageJ software (Version 8.0.2 (263)). Statistical significance between two groups was evaluated with Student’s *t*-test. All data fitting and statistical analysis was performed using GraphPad Prism software (version 8.0.2 (263)). In the figures the criterion for statistical significance was set at * *p* < 0.05, ** *p* < 0.01, and *** *p* < 0.001.

## 3. Results

### 3.1. ACR-Induced Activation of ERK1/2 and p38^MAPK^; Time-Depndence

A time-dependence of ACR-induced activation of the ERK1/2 and p38^MAPK^ inflammatory/apoptotic pathways was determined in PC12 cells. The cells were incubated with 5 mM ACR for 1 or 5 h at 37 °C, washed, lysed, and proteins were separated on SDS-PAGE. Activated MAPKs were recorded in a Western blot analysis monitoring ERK1/2 phosphorylation (Figure 1A) and p38^MAPK^ phosphorylation (Figure 1B), using the corresponding anti-phospho antibodies. As shown, phosphorylation of ERK1 (p-ERK1), ERK2 (p-ERK2), and p38^MAPK^ (p-p38^MAPK^) was significantly elevated at 5 mM ACR after 5 h of incubation of the cells, compared to a small increase observed after 1 h (Figure 1A,B).

### 3.2. ACR-Induced Activation of ERK1/2 and p38^MAPK^; Concentration-Dependence

Next, a more detailed time response and a concentration dependence of ACR-induced p38^MAPK^ activation was tested. The cells were incubated with increasing concentrations of ACR, as indicated, and 5 h later were lysed and proteins were separated on SDS-PAGE. The phosphorylation of p38^MAPK^ was monitored using the corresponding anti-phospho-p38^MAPK^ antibody in a Western blot analysis (Figure 2A). The time-dependence of ACR activation was tested at 1, 3, 4, and 5 h of incubation with 5 mM ACR (Figure 2B). As shown, a robust activation of p38^MAPK^ phosphorylation was observed at 3, 4, and 5 h of incubation, with a slight increase noted after 1 h. There was no significant change in the extent of p38^MAPK^ phosphorylation between 3, 4, and 5 h.

In the following experiments, we investigated the effects of ACR exposure over a 3 h time course. Cells were initially treated with ACR for 2 h, followed by an additional 1 h period after removal of ACR from the medium to assess continued intracellular ACR activity, resulting in a total exposure time of 3 h.

In the experiments testing the effects of the various antioxidants, cells were initially treated with ACR for 2 h and then it was removed. We determined the effects of compounds having a single Cys residue, compared with compounds with two and three Cys residues. We also compared D- to L-isomers and the presence of levodopa flanked by 2 Cys residues. Various compounds were added to the cells for 1 h after 2 h incubation with ACR, and they were then lysed and analyzed for MAPK phosphorylation.

### 3.3. Inhibition of ACR Induced ERK1/2 and p38^MAPK^ Phosphorylation by N-Acetylcysteine (NAC) and N-Acetylcysteine Amide (AD4/NACA)

PC12 cells were incubated with 5 mM ACR for 2 h. Following the removal of ACR from the medium, the cells were treated with or without NAC at the indicated concentrations for an additional hour (Figure 3A,B). The removal of ACR from the medium precluded any interaction between ACR and NAC outside the cells, ensuring that any observed effects were exclusively intracellular. After incubation, the cells were lysed, and proteins were separated using 10% SDS-PAGE. Phospho-ERK1/2 and phospho-JNK were detected using the corresponding anti-phospho antibodies (Figure 3B). Inhibition of ACR-induced activation of the MAPK by NAC was used to calculate an IC_50_ value, a measure of the concentration required to inhibit 50% of ERK1 phosphorylation. The IC_50_ of ERK1 was 131 ± 0.4 µM, the IC_50_ of ERK2 was 133 ± 0.4 µM, and the IC_50_ of JNK was102 ± 22 µM (Figure 3C,D,E; Table 1).

Next, AD4/NACA, a membrane-permeable derivative of NAC, was tested for its ability to inhibit ACR-induced ERK1/2 and JNK under the same experimental conditions (Figure 4A). AD4/NACA has been previously shown to be more potent than NAC in inhibiting auranofin induced ERK1/2 phosphorylation [39,44,45]. As shown, AD4/NACA exhibited a dose dependent inhibition of ACR-induced phospho-ERK1 activation (IC_50_ = 47 ± 0.1 µM), phospho-ERK2 (IC_50_ = 93 ± 0.1 µM), and phospho-JNK1 (IC_50_ = 21 ± 0.7 µM), which had between 2- and 6-fold higher potency compared to NAC (Figure 4B–E; Table 1).

### 3.4. AD4/NACA and AcCys-Pro-CysNH_2_ (TXM-CB3) Prevent ACR-Induced ERK1/2 Activity

AD4/NACA was also tested for its ability to protect the cells prior to ACR administration. The cells were incubated with AD4/NACA at 0.5, 1.0, and 2.0 mM, or with 0.2 mM of the potent antioxidant Ac Cys-Pro-Cys NH_2_, a thioredoxin mimetic peptide called TXM-CB3 [34,35]. After 20 min, AD4/NACA and TXM-CB3 were removed from the cells and 5 mM ACR was added for 3 h incubation. Then, the cells were washed and lysed, and the proteins were separated on SDS-PAGE. ACR-induced phospho-p38^MAPK^ was determined by Western blot analysis and quantified by normalization with β-catenin. A significant reduction in p38^MAPK^ phosphorylation was already observed at 0.5 mM AD4/NACA, and almost full inhibition was observed at 2.0 mM, or at 0.2 mM TXM-CB3 (Figure 4F,G). These results show that pretreatment of the cells with redox thiol-compounds prevents ACR-induced activation of the apoptotic-inflammatory pathway. TXM-CB3 appeared to be ~10-fold more potent than AD4/NACA in preventing ACR toxicity, which can be attributed to its two Cys residues, and is supported by previous in vitro studies [34,35] and in vivo experiments [41,46]. The effect of ACR on cell morphology was visualized in the presence of AD4/NACA in PC12 cells. Following a 2 h incubation period with 5 mM ACR, ACR was removed from the medium, and the cells were treated with or without 1.0 mM AD4/NACA for an additional 1 h (Figure 4H). Phase-contrast microscopy revealed significant morphological alterations in PC12 cells exposed to ACR, exhibiting increased graininess and shrinkage compared to untreated control cells. In contrast, PC12 cells treated with AD4/NACA appeared healthy, displaying morphology similar to that of the untreated control cells. This observation suggests a protective effect of AD4/NACA against ACR-induced toxicity.

### 3.5. Reversal of ACR Toxicity on Cell Viability by NAC, AD4/NACA and TXM-CB3

PC12 cells were treated with 2.5 mM ACR for 2 h, washed and incubated with NAC (0.2–2.0 mM; Figure 5A), AD4/NACA (0.05–1.0 mM; Figure 5B) or TXM-CB3 (5–50 µM Figure 5C). Cell survival was determined after 48 h by the MTT assay. A significant increase in cell survival was observed with 2.0 mM NAC and no effect was observed at 0.5 mM. There was a significant increase in cell viability by AD4/NACA and TXM-CB4 at 0.5 mM. These results are consistent with higher potency of AD4/NACA and TXM-CB3 over NAC in inhibiting ACR-induced MAPK activation.

### 3.6. Inhibition of ACR-Induced ERK1/2 and p38^MAPK^ Phosphorylation by TXM-CB20 (L-Form) and TXM-CB30 (D-Form)

Next, we tested the rescue of cells after being exposed to ACR by the members of the TXM family. These N- and C-blocked tri- (AcC-x-C NH_2_) and tetra-peptides (AcC-x-x-C NH_2_) have previously been shown to be effective in inhibiting oxidative stress-induced MAPK activation in correlation with inhibition of the inflammatory and apoptotic pathways [34,35,38,39,41,42,43,47,48,49].

The tri-peptides TXM-CB20 (Ac Cys-Gly-Cys NH_2_) composed of one Gly and two Cys residues, and TXM-CB30 (Ac DCys-Gly-DCys NH_2_) composed of one Gly and two stereoisomer (DCys) residues, were tested for their ability to reverse ACR-induced MAPK activation (Figure 6). PC12 cells were incubated with 5.0 mM ACR for 2 h, washed, and incubated with increasing concentrations of TXM-CB20 (Figure 6A–E), or TXM-CB30 (Figure 6F–J) for an additional 1 h. Inhibition of ACR-induced activation of the MAPK-apoptotic pathways by either one of the isoforms was evaluated by monitoring the phosphorylation of ERK1/2 and p38^MAPK^, and quantified by normalizing to β-catenin. The proteolysis-resistant D-peptide, which was designed for higher cellular stability demonstrated greater protection against ACR toxicity, exhibiting a 2-fold higher potency in inhibiting ERK2 phosphorylation compared to the corresponding L-peptide, TXM-CB20. Both peptides demonstrated similar inhibition of ACR-induced phosphorylation of p38^MAPK^ (Table 1).

### 3.7. Inhibition of ACR-Induced ERK1/2 and p38^MAPK^ Phosphorylation by the TXM-CB13

Next, we tested the inhibition of ACR-induced MAPK activity by TXM tetra-peptide, Ac Cys-Met-Lys-Cys amide (DY-70; TXM-CB13). PC12 cells were incubated with 5.0 mM ACR for 2 h, followed by an additional 1 h incubation with increasing concentrations of TXM-CB20. As shown in Appendix A, TXM-CB13 appeared to be highly effective in inhibiting ACR-induced activation of ERK1 (IC_50_ = 10.4 ± 4.5 µM), ERK2 (IC_50_ = 7.96 ± 3.6 µM), and p38^MAPK^ (IC_50_ = 12.4 ± 1.3 µM) (Table 1). These results are consistent with TXM-CB13’s inhibition of auranofin-induced p38^MAPK^ and JNK phosphorylation in human neuronal SH-SY5Y cells [47], as well as inhibition of platelet aggregation induced by collagen [50].

TXM-CB13 has also been shown to be highly effective in vivo, improving impaired cognitive performance after mild traumatic brain injury (mTBI) [38]. The higher efficacy demonstrated by the TXM tetra-peptide compared to TXM tri-peptides is also consistent with findings from earlier in vivo and in vitro studies [34,38,51].

The activation of cyclic AMP response element-binding protein (CREB) by ACR was effectively inhibited by TXM peptides, specifically TXM-CB13 and TXM-CB3 (Appendix A). CREB is a crucial transcription factor involved in cell survival and synaptic plasticity, and its activation is often linked to cellular stress responses and neurotoxic effects The inhibition of CREB phosphorylation was used as a measure of the efficacy of TXM peptides in countering ACR-induced inflammatory signaling pathways. We also tested two peptides for their ability to inhibit CREB phosphorylation downstream to ERK1/2 and ribosomal S6 kinase (RSK). The IC_50_ values of TXM-CB13 IC_50_ = 4.7 ± 7.5 µM (Appendix A) and TXM-CB3 IC_50_ = 28.9 ± 6.7 µM (Appendix A), were determined using anti-phospho-CREB antibodies. Again, the tetra-TXM peptide displayed higher efficacy as compared to the tri-peptide.

### 3.8. Inhibition of ACR-Induced ERK1/2 and p38^MAPK^ Phosphorylation by TXM-CB16, a 3-Cys Residues Peptide

We tested inhibition of ACR-induced MAPK activation by AcCys-γGlu-Cys-Cys NH_2_ (TXM-CB16). This tetra-peptide, in addition to having Cys residues in position 1 and 4, contains a third Cys residue at position 2 and a γ-glutamyl residue [47]. The unique structure suggests that proteolysis can produce γ Glu-Cys, a dipeptide that is a rate limiting substrate of GSH (γ-Glu-Cys-Gly) and by itself is a strong antioxidant (Figure 7). PC12 cells were incubated with 5 mM ACR for 2 h followed by an additional 1 h incubation with increasing concentrations of TXM-CB16, as indicated. After lysis and protein separation on SDS-PAGE we determined the phosphorylation of ERK1/2 and p38^MAPK^ using the corresponding anti-phospho-ERK1/2 antibodies. As shown in Figure 8, TXM-CB16 effectively inhibits ACR-induced ERK2 (IC_50_ = 9.0 ± 2.5) and p38^MAPK^ activation (IC_50_ = 14.8 ± 1.7 µM) (Table 1).

### 3.9. Inhibition of ACR-Induced ERK1/2 and p38^MAPK^ Phosphorylation by Dopamine Precursor SuperDopa (SD)

In light of the Parkinsonian-like effects observed after chronic ACR exposure, we tested SuperDopa (SD), a TXM peptide designed to act as an antioxidant and provide the cell with levodopa [40,52]. PC12 cells were incubated with 5 mM ACR for 2 h, washed, and incubated with SD for additional 1h at increasing concentrations, as indicated. The cells were lysed, and proteins were separated on 10% SDS-PAGE. ERK1/2 and p38^MAPK^ phosphorylation were determined using the corresponding anti-phospho-ERK1/2 or anti-phospho-p38^MAPK^ antibodies. As shown in Figure 9, SD effectively inhibits the ACR-induced phosphorylation of ERK1/2, displaying an IC_50_ of 4.7 ± 0.7 µM, and p38^MAPK^ phosphorylation with an IC_50_ of 34 ± 5.5 µM (Table 1).

## 4. Discussion

Acute and chronic ACR toxicity, along with its potential hazard in industry spills, has led us to explore promising antidotes. ACR is known to elevate ROS and to irreversibly modify Cys residues, e.g., GSH and selective proteins such as EGF receptors [53]. Here, we have shown that thiol-based molecules reverse activation of MAPK signaling pathways and may act as potential antidotes to ACR-induced toxicity. The two-fold inhibition appeared to be correlated with the number of Cys residues, and could be attributed to ROS scavengers, or to ACR targeting selective kinases and phosphatases [39].

### 4.1. ACR-Induced Activation of Intracellular Pathways

We have shown that ACR induces a significant increase in the phosphorylation of ERK1/2, and p38^MAPK^. The elevated activity, which could be attributed to covalent modification by ACR of an essential Cys residue of the MAPKs or decreased activity by the corresponding phosphatases, appeared maximal already at 3 h incubation with 5 mM ACR. Activation showed a marginal increase after 1 or 2 h, and no significant activation was observed at 1 or 2 mM ACR.

When the cells were exposed to 5 mM for 30 min and allowed to incubate for 5 h, no ACR-induced activation was detected, suggesting that ACR entry over 30 min is not sufficient to cause MAPK activation. 

### 4.2. Reversal of ACR-Induced MAPK Activation

Two different approaches were applied to test the potential ACR inhibitory efficacy of the thiol-based compounds that can freely cross the cell membrane. A “prevention” approach by administrating the thiol compound prior to ACR, and a “treatment” approach, in which the thiol compounds were administrated subsequent to ACR exposure.

Most of our studies are focused on the “treatment” approach, for developing a potential future treatment of ACR toxicity, in which the effects of the thiol compounds were applied to cells pre-exposed to ACR.

In the “treatment” approach, the cells were incubated for 2 h, washed, and then exposed for an additional 1 h to the tested compounds. Although ACR is washed out of the medium after 2 h, it remains inside the cells for a total of 3 h, which remains high until 5 h. Accordingly, our results monitor the ability of the thiol-compounds to reverse the toxicity of ACR that has penetrated the cells. In both the “preventive” and the “treatment” assays, the thiol compounds and ACR were not present together in solution, preventing direct interaction between them.

In the “treatment” and the “preventive” assay, two types of thiol-based antioxidants were tested, the single Cys residue antioxidants, NAC or AD4/NACA, and the TXM peptides, comprising either 2 Cys or 3 Cys residues. The single Cys-molecules NAC and AD4/NACA, and the 2 Cys TXM-peptides, whose structure is based on the redox motif of thioredoxin (CxC or CxxC), and has been shown to mimic thioredoxin redox activity, effectively reverse oxidative-stress induced MAPK activation (review [39]).

The dose-dependent inhibition of ACR-induced inflammatory activity by AD4/NACA revealed a considerably higher potency compared to NAC. This higher potency of AD4/NACA compared to NAC has previously been observed in reversing auranofin-induced MAPK phosphorylation, and was attributed to the increased membrane permeability of AD4/NACA when compared to membrane-impermeable NAC [39,44]. These results are consistent with a study in which NAC provided negligible protection in ACR-induced acute neurotoxicity. In that study, failure of NAC to protect was attributed to both low BBB permeability and high deacetylation of NAC during intestinal absorption [54].

### 4.3. Inhibition of ACR-Induced MAPK Activation by TXM-Peptides

As compared to the single Cys antioxidants NAC and AD4/NACA, the tri-peptide (TXM-CB20) and its stereoisomer (TXM-CB30) demonstrated a higher potency in inhibiting ACR-induced activation of p38^MAPK^ and ERK2 (Table 1). These results are consistent with earlier studies indicating a higher redox efficacy of the dithiol tri- and tetra-TXM-peptides at lowering, auranofin-induced MAPK activation, compared to single Cys compounds [39]. Moreover, the proteolysis-resistant TXM-CB30 displayed a 2-fold higher efficacy in inhibiting ACR-induced ERK2 activity compared to TXM-CB20. This higher potency over the corresponding hydrolysable L-peptide is likely due to its resistance to proteolysis. The advantages of being a proteolysis-resistant peptide should manifest itself in vivo, in prolonged redox activity.

Although the inhibition of ACR-induced MAPK activity by the tri-TXM-peptides was higher compared to NAC or AD4/NACA, it was lower compared to the tetra-peptide TXM-CB13. This higher potency parallels the higher effectiveness of TXM tetra-peptides in reducing MAPK activation induced by auranofin, as demonstrated previously with oxidative stress-induced MAPK [34,51]. Also, in vivo, TXM-CB13 appeared to be considerably more potent than NAC or AD4/NACA in protecting cognitive activity after mild traumatic brain injury (mTBI) [38]. ACR-induced CREB activation, which indicates activation of the ERK/RSK/CREB transcription pathway, is inhibited more potently by TXM-CB13 compared to TXM-CB3. This difference further substantiates the higher efficacy of the tetra-TXM-peptide as compared to the TXM tri-peptide in protecting the cells from oxidative stress and inflammatory stress. A recent study showed an increase in phospho-CREB in SH-SY5Y cells subjected to 70 µM ACR after 9 days, consistent with ACR interference with signaling pathways. In this study, the expression of 18 out of 47 genes in the CREB signaling pathway were altered, revealing a decrease in neurite outgrowth and involvement in neuronal differentiation [55]. The results of the cell survival MTT assay are consistent with higher potency of AD4/NACA and TXM-CB3 over NAC in inhibiting ACR-induced MAPK activation.

### 4.4. Inhibition of ACR-Induced MAPK by TXM-CB16, a 3-Cys Tetra-TXM Peptide

Based on results demonstrating that dithiol TXM peptides exhibit greater potency compared to single Cys peptides and that tetra-TXM peptides display higher potency than tri-TXM peptides, we further examined TXM-CB16 (DY-71) a tetra-peptide containing three Cys residues. Potentially, upon cell entry and proteolytic cleavage, TXM-CB16 may generate fourteen distinct Cys-containing peptides. The TXM-CB16 peptide has been shown to be effective in lowering insulin- and oxidative stress-induced ERK1/2 activation, rescuing HepG2 cells from cell death [47]. In addition to having three Cys residues and being a TXM-peptides having the thioredoxin -CxxC- motif, TXM-CB16 has a γGlu residue, which is a substrate of GSH synthetase (Figure 7). Despite these properties, and its potential ability to provide the cell with γGluCys, which by itself is a potent antioxidant, TXM-CB16 displayed similar potency compared to the dithiol-based tetra-peptide TXM-CB13 (Table 1). It is possible that longer assay conditions are needed to provide sufficient time for peptide proteolysis and might show higher efficiency in vivo.

### 4.5. Inhibition of ACR-Induced MAPK Activation by SuperDopa, a Dithiol Levodopa Peptide

The PD-like symptoms displayed by individuals exposed to ACR indicate possible damage in dopaminergic neurons. In rats, chronic exposure (12 months) to ACR caused motor dysfunction, α-synuclein accumulation, and decreased brain-derived neurotrophic factor (BDNF). These hallmarks of Parkinson’s disease were correlated with depletion of dopaminergic neurons in the substantia nigra [13]. Efforts to protect dopaminergic neurons have shown the efficacy of AD4/NACA in decreasing damage in three experimental models of PD, the 6-OHDA-induced nigral lesion, the 1-methyl-4-phenyl-1,2,3,6,-tetrahydropyridine (MPTP), and a chronic intrajugular administration of rotenone in rats [56]. More recently, SuperDopa (SD) has been shown to protect the motor activity of rotenone-treated rats and inhibit auranofin-induced activation of the apoptotic pathway in SH-SY5Y cells [40]. SD protects ARPE-19 cells from photic and non-photic stress, as well as from Rose Bengal photosensitization-induced oxidation [52]; see also [57]. In the present study, SD-like TXM-CB13 and TXM-CB16 displayed higher efficacy in reversing ACR-induced MAPK activation compared to the single Cys residue AD4/NACA or NAC. This activity arises predominantly from having two Cys residues, ultimately leading to a reduction in oxidative stress through scavenging free radical oxidation [52]. However, SD like TXM-CB13 and TXM-CB16 may offer additional benefits in vivo through multiple Cys-containing peptide fragments generated during prolonged exposure within the cell, e.g., potential fragmentation of SD [40]. Moreover, given that SD may also generate levodopa, it might contribute to the replenishment of dopamine levels within dopaminergic neurons, alleviating Parkinson’s-like effects.

#### Limitation of the Study

Unlike chronic toxicity studies that focus on micromolar concentrations of ACR typically found in food, this study examines the effects of acute exposure to higher concentrations of ACR. The scope is confined to the three MAPK pathways and does not extend to other ACR-modified cellular processes. It remains important to determine whether the thiol-based antioxidants directly interact with or scavenge ACR to prevent ROS production, or if their effects on MAPKs are due to scavenging a secondary increase in ROS.

## 5. Conclusions

In our efforts to prevent the toxicity associated with ACR-induced activation of the MAPK pathways, we developed a reproducible assay designed to test thiol-compounds for reversing ACR activation of MAPK signaling. Using this assay, we identified several selective TXM peptides, namely, TXM-CB13, TXB-CB30, TXM-CB16, and SD, that exhibit promising efficacy in suppressing ACR-induced activation of the MAPK pathways. The inhibition demonstrated a dose-dependent response, with potency correlating directly with the number of Cys residues present. TXM peptides were considerably more effective compared NAC or AD4/NACA. SD, one of the TXM-peptides containing a levodopa residue could become effective in alleviating Parkinson’s disease-like symptoms following acute ACR exposure, not only through lowering oxidative stress, but also by replenishing cellular levels of levodopa. Furthermore, by saturating the cell with free-thiol groups, we can rescue GSH, thioredoxin, glutaredoxin, and other thiol-dependent redox enzymes from covalent inactivation by ACR. Altogether, these findings underscore the potential of TXM peptides as effective ACR-antidote candidates, inhibiting oxidative stress-related neurotoxicity and inflammation.

## Figures and Tables

**Figure 1 antioxidants-13-01431-f001:**
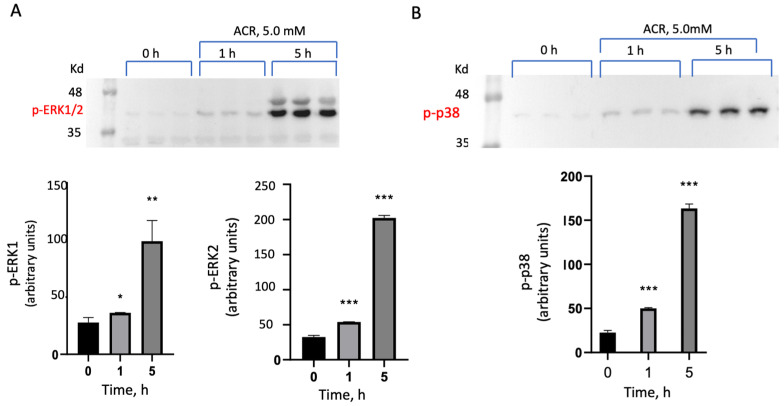
Time course of ACR-induced ERK1/2 and p38^MAPK^ activation. PC12 cells were incubated with 5mM ACR for 1 and 5h. Then, the cells were washed, lysed, and proteins were separated on SDS-PAGE and analyzed by Western blot analysis. MAPK phosphorylation of (**A**) ERK1/2 and (**B**) p38^MAPK^. The results were quantified and presented as arbitrary units of normalized p-ERK1/2 and pp38^MAPK^ expression using the corresponding anti-phospho-ERK1/2 antibodies and the anti-phospho-p38^MAPK^ antibodies. * *p* < 0.05; ** *p* < 0.01; *** *p* < 0.001.

**Figure 2 antioxidants-13-01431-f002:**
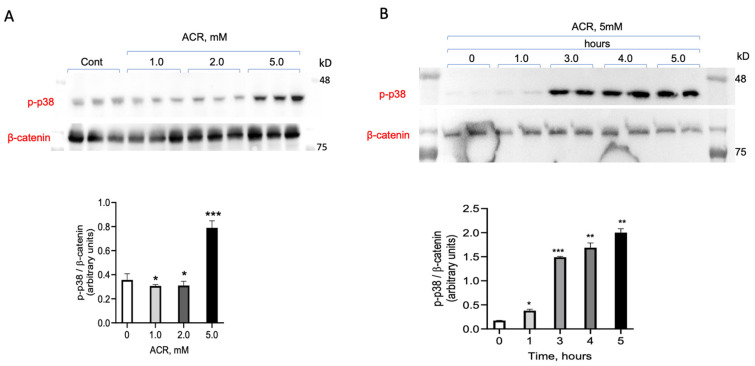
Concentration dependency and time course of ACR-induced p38^MAPK^ activation. (**A**) PC12 cells incubated with increasing concentrations of ACR (1, 2, and 5 mM) for 5 h. Equal protein amounts of cell lysates separated on 10% SDS-PAGE and ACR-induced activation of p38^MAPK^ was monitored and analyzed by Western blot using anti-phospho-p38^MAPK^ antibodies. The results were quantified and presented as arbitrary units of phopsho-p38^MAPK^ expression normalized to β catenin (**B**) PC12 cells incubated with 5 mM ACR for 1, 3, 4, and 5 h. ACR-induced activation of p38^MAPK^ was monitored by Western blot analysis as in (**A**). The values shown are averages (±SEM) of two independent experiments (Materials and methods) * *p* < 0.05; ** *p* < 0.01; *** *p* < 0.001.

**Figure 3 antioxidants-13-01431-f003:**
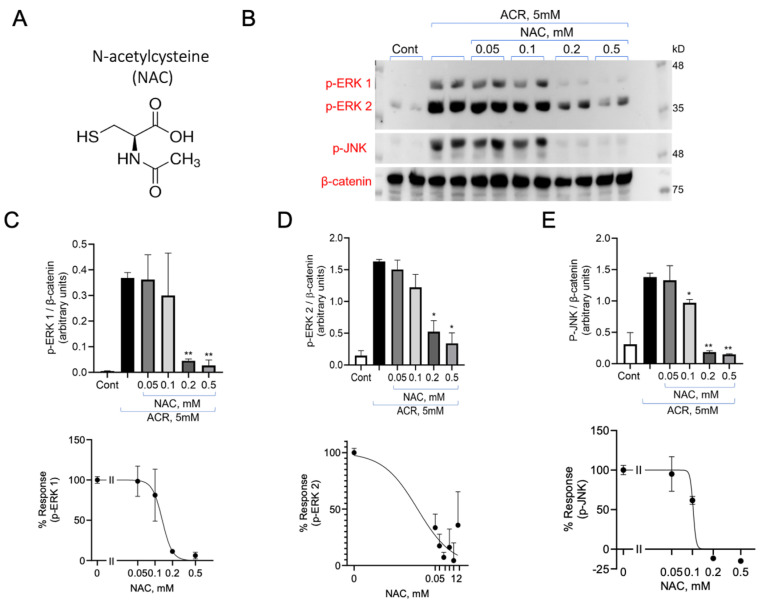
NAC inhibits ACR-induced activation of the MAPKs. (**A**) The structure of NAC. (**B**) PC12 cells were treated with 5 mM ACR for 2 h and incubated with or without increasing concentrations of NAC, as indicated (in duplicates). Equal amounts of cell lysate protein were separated on 10% SDS-PAGE and analyzed by immunoblotting with the corresponding antibodies. The results were quantified and presented as arbitrary units of expression. (**C**) p-ERK1, (**D**) pERK2, (**E**) pJNK1 normalized to β catenin. The values shown are averages (±SEM) of three independent experiments in duplicate. Student’s *t*-test (two populations) was performed for ACR treated cells. * *p* <0.05; ** *p* value < 0.01.

**Figure 4 antioxidants-13-01431-f004:**
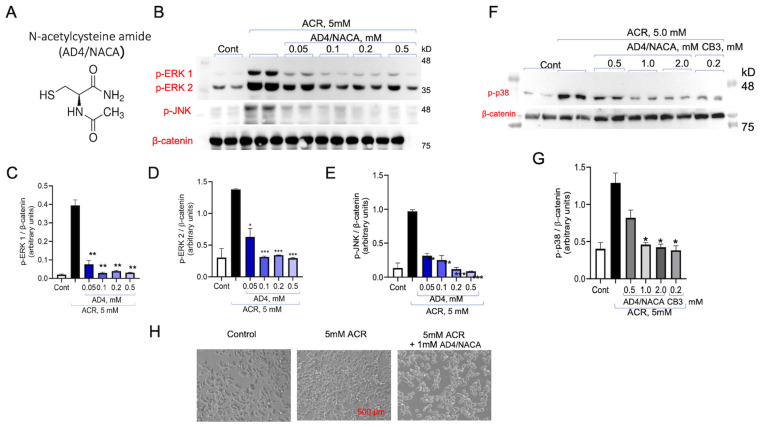
AD4/NACA inhibits ACR-induced activation of MAPKs. (**A**) The structure of AD4/NACA. (**B**) PC12 cells treated with 5 mM ACR for 2 h, washed and incubated with increasing concentrations of AD4/NACA. Proteins of cell lysates were separated on 10% SDS-PAGE and phosphorylation of (**C**) pERK1, (**D**) pERK2, (**E**) pJNK, and p38^MAPK^ (**F**,**G**) were normalized to β catenin. The values shown are averages (±SEM) of two independent experiments in duplicate, normalized to the phosphorylation state of cells treated with ACR after 3 h. Student’s *t*-test (two populations) was performed for ACR treated cells: * *p* < 0.05; ** *p* <0.01; *** *p* < 0.001. (**H**) Phase-microscope images of PC12 cells taken after 24 h incubation with 5 mM ACR or 5 mM ACR with 1 mM AD4/NACA.

**Figure 5 antioxidants-13-01431-f005:**
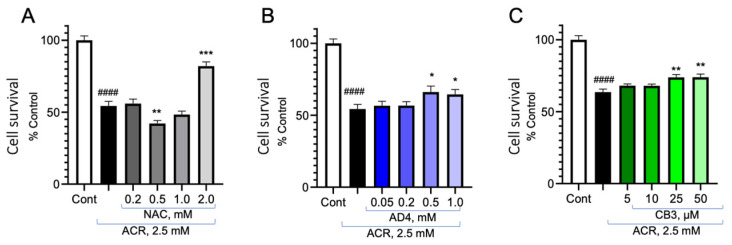
Reversal of ACR-induced cell death by NAC, AD4/NACA, and TXM-CB3. PC12 cells in a 96-well plate were incubated with 2.5 mM ACR for 2 h, and after washing, were incubated with or without (**A**) NAC, (**B**) AD4/NACA, and (**C**) TXM-CB3 for 48 h at concentrations, as indicated. Cell viability was determined using the MTT assay. The data are expressed as a mean of octuplicates ± SEM. #### Control vs. 2.5 mM ACR. * *p* < 0.05, ** *p* < 0.01, *** *p* < 0.001.

**Figure 6 antioxidants-13-01431-f006:**
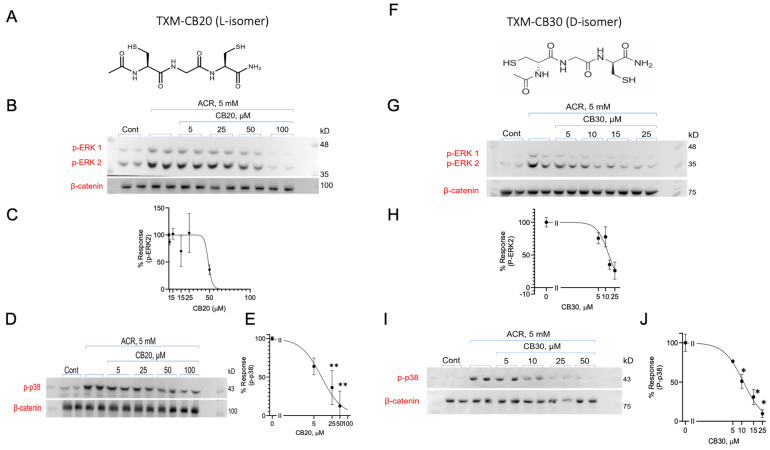
TXM-CB20 and TXM-CB30 inhibit ACR-induced activation of MAPKs. (**A**–**E**) TXM-CB20 or (**F**,**G**) TXM-CB30. PC12 cells treated with 5 mM ACR for 2 h, washed, incubated with increasing concentrations of TXM-CB20 and protein lysates were separated on 10% SDS-PAGE. Inhibition of ACR-induced phosphorylation of (**C**,**H**) ERK2 and (**D**,**E**,**I**,**J**) p38^MAPK^ was calculated and quantified by normalization to β-catenin. The values shown are averages (±SEM) of two independent experiments in duplicates normalized to β-catenin. Student’s *t*-test (two populations) was performed for ACR treated cells * *p* < 0.05; ** *p* < 0.01.

**Figure 7 antioxidants-13-01431-f007:**
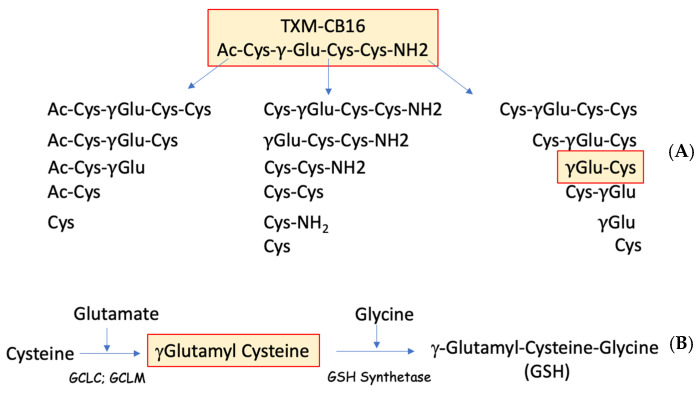
A cluster of thiol-containing compounds is generated by the proteolytic cleavage of TXM-CB16. (**A**) Potentially, upon cell entry and proteolytic cleavage, TXM-CB16 generates 14 distinct cysteine (Cys)-containing peptides. Among these is γ-glutamylcysteine (γGlu-Cys) (**B**) γGlu-Cys serves as a substrate for glutathione (GSH) biosynthesis.

**Figure 8 antioxidants-13-01431-f008:**
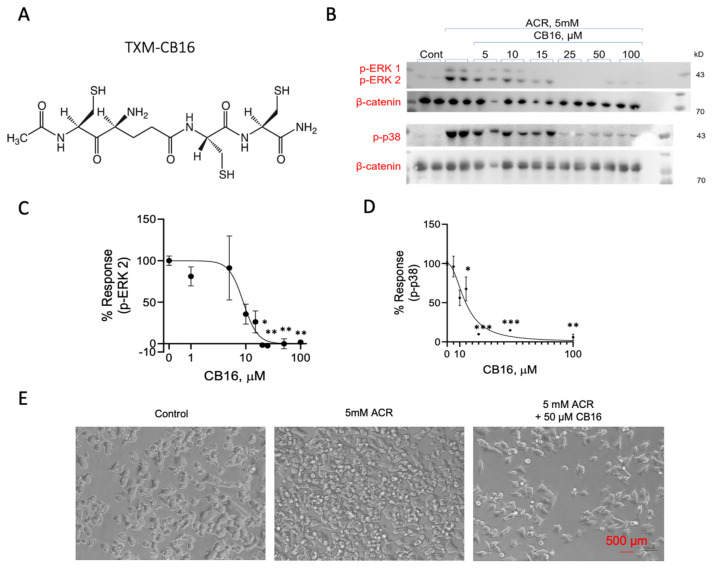
TXM-CB16 inhibits ACR-induced activation of ERK1/2 and p38^MAPK^. (**A**) The structure of TXM-CB16. (**B**) PC12 cells were treated with 5 mM ACR, washed for 2 h, and then treated with increasing concentrations of TXM-CB16. Lysate proteins were separated on 10% SDS-PAGE. Inhibition of ACR-induced phosphorylation of (**C**) ERK2 and (**D**) p38^MAPK^ was calculated and quantified. The values shown are averages (±SEM) of two independent experiments normalized to β-catenin, with the phosphorylation state of cells treated with ACR after 3 h. (**E**) Phase-microscope images of PC12 cells taken after 24 h incubation with 5 mM ACR or 5 mM ACR with 50 µM TXM-CB16. Student’s *t*-test (two populations) was performed for ACR treated cells. * *p* < 0.05; ** *p* < 0.01; *** *p* < 0.001.

**Figure 9 antioxidants-13-01431-f009:**
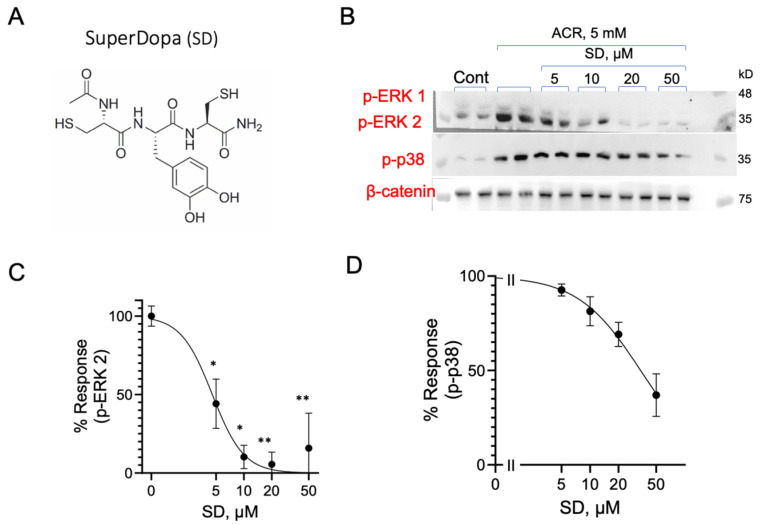
SuperDopa (SD) reverses ACR-induced activation of ERK1/2 and p38^MAPK^. (**A**) The structure of SuperDopa (SD). (**B**) PC12 cells treated with 5 mM ACR for 2 h, washed, and then incubated with increasing concentrations of SD. Inhibition of ACR-induced phosphorylation of (**C**) ERK2 and (**D**) p38^MAPK^ was quantified, and the values shown are averages (±SEM) of two independent experiments normalized to β-catenin compared to phosphorylation state of cells treated with ACR after 3 h. * *p* < 0.05; ** *p* < 0.01.

**Table 1 antioxidants-13-01431-t001:** Potency of the various thiol-based antioxidants in protecting PC12 cells from ACR-induced MAPK activation.

		* IC_50_, µM			
NAME	STRUCTURE	p-ERK 1	p-ERK 2	p-p38	pJNK
NAC	Ac **Cys**	131 ± 0.4	133 ± 0.4	ND	102 ± 22
AD4/NACA	Ac **Cys**-NH_2_	47 ± 0.1	63 ± 0.1	ND	21 ± 0.7
CB20	Ac **Cys**-Gly-**Cys** NH_2_	97.0 ± 23	48.6 ± 11	9.8 ± 6.8	3.2 ± 1.7
CB30	Ac **DCys**-Gly-**DCys** NH_2_	3.06 ± 1.2	17.9 ± 4.8	11.5 ± 1.9	ND
CB13	Ac **Cys**-Met-Lys-**Cys** NH_2_	10.4 ± 4.5	8.0 ± 3.6	12.4 ± 1.3	ND
CB16	Ac **Cys**-γGlu-**Cys-Cys** NH_2_	ND	9.01 ± 2.5	14.8 ± 1.7	ND
SD	Ac **Cys**-Levodopa-**Cys** NH_2_	4.7 ± 0.7	4.55 ± 0.4	34.2 ± 5.5	ND

* IC_50_—the effective half concentration based on 50% inhibition of phosphorylation of ERK1, ERK2, p38^MAPK^, and JNK.

## Data Availability

All relevant data are within the paper.

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
