# Peer review of "Thiol-Based Redox Molecules: Potential Antidotes for Acrylamide Toxicity"

_antioxidants, 2024, doi:10.3390/antiox13121431_

Round 1
Reviewer 1 Report
Overall this is an interesting study that shows that treatment of cells with thiol-based antioxidants after acrylamide (ACR) administration may be a suitable means of mitigating its toxicity. It would be suitable for publication in Antioxidants if it was more mechanistic.
My major concern is that the only real readout of ACR toxicity is MAP kinase phosphorylation and its attenuation. It would be better to do further downstream studies - for instance, actually looking at cell death (which should be fairly straightforward to do). This need not be for all molecules (maybe pick the most potent ones and compare to NAC), but I think something deeper than just looking at MAPK phosphorylation states is critical. The images of microscopic images of cells are not really that informative or useful, so those could potentially be replaced with quantitative cell death measurements. Also, it might be appropriate to try to determine if the thiol-based antioxidants are directly reacting with/scavenging ACR or if the effects on MAPKs are due to a secondary increase in ROS (which they could also scavenge). While there's not an ideal way to do this experiment, it could potentially be done by administering catalase to the media instead of the thiol-based antioxidants and monitoring effects on MAPK phosphorylation/cell death.
From the standpoint of readability, the figures become a little monotonous and repetitive, since almost all measure the same thing with a different thiol-based antioxidant. It might be best to show one or two of these, and summarize the other molecules in a well-organized table and in supplementary material. In this table, IC50s for toxicity could also be included. Likewise, I don't think there is a need for two different graphs of every single blot shown; my preference would be to show the log-scale graph where the IC50 is calculated.
There are several typographical errors in the text. A few that I caught...
•line 45-46: "significantly small" should be "low"
•line 175: "dependent" should be "dependence"
•line 198: "remove" should be "removed"
•line 199: "Cy" should be "Cys"
Since the manuscript requires more significant revision, this is just a partial list.
Last, the figures could be bigger in size to make them more legible. There's plenty of room to expand them within the margins.
Author Response
Reviewer 1
1Major comments
Overall this is an interesting study that shows that treatment of cells with thiol-based antioxidants after acrylamide (ACR) administration may be a suitable means of mitigating its toxicity. It would be suitable for publication in Antioxidants if it was more mechanistic.
My major concern is that the only real readout of ACR toxicity is MAP kinase phosphorylation and its attenuation. It would be better to do further downstream studies - for instance, actually looking at cell death (which should be fairly straightforward to do).
This need not be for all molecules (maybe pick the most potent ones and compare to NAC), but I think something deeper than just looking at MAPK phosphorylation states is critical. The images of microscopic images of cells are not really that informative or useful, so those could potentially be replaced with quantitative cell death measurements.
Reply
Thank you for pointing this out. According to the reviewer suggestion we inserted our data related to ACR-induced cell death and the effect of several antioxidants on cell survival See new Fig.5
Also, it might be appropriate to try to determine if the thiol-based antioxidants are directly reacting with/scavenging ACR or if the effects on MAPKs are due to a secondary increase in ROS (which they could also scavenge). While there's not an ideal way to do this experiment, it could potentially be done by administering catalase to the media instead of the thiol-based antioxidants and monitoring effects on MAPK phosphorylation/cell death.
From the standpoint of readability, the figures become a little monotonous and repetitive, since almost all measure the same thing with a different thiol-based antioxidant.
Reply
We modified the figures and transferred to the supplementary material as suggested by the reviewer. We combined Fig. 4 and Fig. 5, (new Fig. 4) and Fig. 6 with Fig.7 `were combined into one new Fig. 6. We added a new Fig.% with new data ,according to the reviewer’s re suggestion. Fig. 9 is now fig. 7 and Fig 10 is now Fig.9. The old Fig.8 was transferred to the supplementary material as suggested by the reviewer, and is now Fig. S1. The older Fig. S1 is now Fig.S2
It might be best to show one or two of these, and summarize the other molecules in a well-organized table and in supplementary material. In this table, IC50s for toxicity could also be included. Likewise, I don't think there is a need for two different graphs of every single blot shown; my preference would be to show the log-scale graph where the IC50 is calculated.
Reply
We took the reviewer suggestion and showed only the log-scale graphs, where the IC50 was calculated. We inserted the relevant IC50 in the Table and rearranged the figures. As suggested, we also transferred one figure to the supplementary material and combined two figures into one
Detail comments
There are several typographical errors in the text. A few that I caught...
- line 45-46: "significantly small" should be "low" was corrected
- line 175: "dependent" should be "dependence" was corrected
- line 198: "remove" should be "removed" was corrected
- line 199: "Cy" should be "Cys" was corrected
Since the manuscript requires more significant revision, this is just a partial list.
Last, the figures could be bigger in size to make them more legible. There's plenty of room to expand them within the margins.
Reply
According to the reviewer request we made the figures bigger
Reviewer 2
The authors performed a solid study on the protective effects of certain thiols against acrylamide cytotoxicity, the later expressed as the suppression of its action on a number of cell signaling proteins. However, the study possesses some shortcomings which limits its significance (detailed below).
Detail comments
The main problematic point is too high (millimolar) concentrations of acrylamide used in this study, whereas its content in baked foods may approach 10(-5) M.
Instead, a chronic toxicity study at its micromolar concentrations is recommended, with parallel involvement of thiol compounds. The data on the reaction rates of acrylamide with the examined thiols can be obtained. This would facilitate the characterization of the protective effects - is this a result of the direct (chemical) reactions, or the repair of proteins/DNA after their exposure to ROS or formation of covalent adducts with acrylamide.
We agree with the reviewer that in baked foods ACR is present at micromolar rang concentrations, and for reversing these effects one should design long-term experiments that require a chronic approach.
However, the present study was aimed to reverse ACR toxicity that occurs during a large industrial spill. We addressed the possibilities for overcoming toxicity caused by large amounts of ACR, which would be relevant to overcoming acute industrial disasters, including terror attacks
Besides, the authors should discuss the difference between their current and previous [39] data.
According to the reviewer request we added to the discussion the diffrenec between our current and previous data. [[[See The data published in our earlier study described the potency of NAC, AD4 and TXM peptides in reversing oxidative stress induced by inhibiting thioredoxin reductase. The present study shows first that ACR, activates three different MAPK pathways that can be inhibited by TXM-peptides and to a lesser extent by AD4/NACA and NAC. these antioxidinats appear to inhibit ACR-induced activation of the MAPK pathways, may indicate suggest that ACR generates oxidative stress through generating Michael adducts with different redox proteins including thioredoxin and GSH.
Specific and minor points:
Abstract: #19 - disulfide-reducing; was corrected
Introduction: #40 - commas; was corrected
#65 - what is ER? Endoplasmic reticulum, was inserted
#66-82 - the text is simply descriptive, may be condensed; the text was condensed, as suggested #84 - carbohydrates; was corrected
Results: Unclear meaning of #199-202;
The text frequently duplicates the legends to the figures, e.g. Figs. 2,3,5. We omitted the duplicate sentences
Besides, the same phrases are used in several subsections for the description of the treatment. Figures 4F and 10E (morphology) may be shown together comprising separate figure. The data on the effects on some downstream proteins (#351-355) are not analyzed in the Discussion.
Discussion: this section is mostly descriptive (repetition of the text of Results section) with little or too well-hidden additional value.
Reviewer 3
comments
The authors tested the ability of single cysteine, double cysteine, triple cysteine, and triple cysteine Levodopa-containing peptide antioxidants for their ability to prevent MAPK activation in response to acrylamide administration to cell lines in culture. The experimental design is good and the paper is fairly well-written. The data analysis and statistical approach analysis is appropriate. However, I have two major concerns.
Major comments:
Comment 1: The major flaw of the paper is that the authors appear to assume that MAPK activation is the main mechanism of toxicity of ACR in the PC12 cells without showing data or referencing a paper showing this. If no reference is available, the authors need to show that a MAPK inhibitor will decrease the toxicity of ACR on the PC12 cells as the first experiment in the Results section. If the authors cannot show this, their use of PC12 cells is not relevant to the mechanism of ACR toxicity.
Reply
Please note that ACR-induced neurotoxicity through the activation of the MAPK has been shown in several studies including PC12 cells. We have included references to these studies in the introduction; Please see references # 13-18
Also, in vivo, oral administering of ACR to rats through gastric gavage showed a significant effect on p38MAPK ; Please see reference “Environ Sci Pollut Res Int . 2021 Sep;28(36):49808-49819. doi: 10.1007/s11356-021-14049-4 “ /
We do not suggest that the primary mechanism of ACR toxicity occurs through MAPK inhibition. Instead, we investigated whether ACR-induced activation of MAPK pathways, known to initiate inflammatory apoptotic cell death, could be inhibited by thiol-based compounds. Our findings demonstrate that various thiol reagents, which are recognized for their ability to inhibit MAPK activity and prevent cell death, also reduce ACR-induced MAPK activation in PC12 cells
Comment 2: The second largest flaw in the manuscript is the use of PC12 cells to model the toxicity of ACR to substantia nigra neurons in Parkinson’s disease. A dopaminergic cell line should be used in addition to the preliminary results with PC12 cells that directly compare the effects of SuperDopa to the other sulfhydryl-containing peptides.
Reply
I completely agree with the reviewer’s comment that an additional neuronal-like cell line would be more appropriate for modeling Parkinson's disease. However, it is important to note that we do not use PC12 cells as a model for assessing ACR toxicity in substantia nigra neurons related to Parkinson’s disease. Rather, we mention that ACR, which has been associated with Parkinsonian symptoms, may have an additional advantage in case of acute ACR toxicity by the use of SD, which releases levodopa during proteolysis
Comment 3: Limitations of the study should be clearly stated at the end of the Discussion.
Reply
According to the reviewee suggestion we jave added a section of limitations of the study
Limitation of the study
Unlike chronic toxicity studies that focus on the micromolar concentrations of ACR typically found in food, this study examines the effects of acute exposure to higher concentrations of ACR. The scope is confined to the three MAPK pathways and does not extend to other ACR-modified cellular processes. It remains important to determine whether the thiol-based antioxidants directly interact with or scavenge ACR to prevent ROS production, or if their effects on MAPKs are due to scavenging a secondary increase in ROS
Detail comments :
Reply
We are really thankful for the excellent corrections made by the reviewer.
We have corrected all of them.
Line 98: auranofin induced -> auranofin (thioredoxin reductase inhibitor)-induced
Line 175: dependent -> dependence
Line 198: remove -> it was removed
Line 199: Cy -> Cys
Line 200: in flanked -> flanked
Line 200: residues,. -> residues. [remove comma]
Line 213: ERK1 phosphorylation IC50 of 131 ± 0.4 μM, ERK2 IC50 of 133 ± 0.4 μM, and JNK IC50 of 102 ± 22 μM -> The IC50 for ERK1 was 131 ± 0.4 μM, the IC50 of ERK2 was 133 ± 0.4 μM, and the IC50 of JNK was 102 ± 22 μM
Figure 3 panel A: Acetyl-cysteine -> N-acetylcysteine
Line 218: Cell lysates proteins in equal amounts -> Equal amounts of cell lysate protein
Line 222: **P value -> **p
Figure 4A: Acetylcysteine amide -> N-acetylcysteine amide
Line 263: 0.2mM -> 0.2 mM
Line 283: (AcC-x-x-C 282 NH2). -> (AcC-x-x-C 282 NH2) [remove period]
Line 303: ERK1and -> ERK1 and
Line 353: RSK -> ribosomal S6 kinase (RSK)
Line 405: antidots -> antidotes
Line 406: residues -> residues,
Line 407: molecule -> molecules
Line 407: antidot -> antidotes
Line 409: two fold -> two-fold
Line 417: or -> or decreased activity
Lines 418 and 420: 5mM -> 5 mM [Be consistent with a space in between number and mM throughout manuscript]
Line 426: thiol based -> thiol-based
Line 445: ACR induced -> ACR-induced
Line 446: the NAC -> NAC
Line 448: MAPKs -> MAPK phosphorylation
Line 448: membrane -> increased membrane
Line 449: opposed to the membrane impermeable -> when compared to the lower membrane permeability of
Line 450: ACR -> ACR-induced
Line 468: stress induced -> stress-induced
Line 470: Traumatic Brain Injury -> traumatic brain injury
Line 486: TXM-peptides -> TXM-peptide
Line 495: damaging -> damage
Line 498: neuron -> neurons
Line 507: photosensitized induced -> photosensitization-induced
Lines 508 and 512: SD like -> SD-like
Line 519: ACR -> ACR-induced
Line 522: peptides-namely, -> peptides, namely
Line 531: thiol dependent -> thiol-dependent
Line 533: ACR-antidotes -> ACR antidote
Line 535: PD like -> PD-like

Reviewer 2 Report
The authors performed a solid study on the protective effects of certain thiols against acrylamide cytotoxicity, the later expressed as the suppression of its action on a number of cell signaling proteins. However, the study possesses some shortcomings which limits its significance (detailed below).
The main problematic point is too high (millimolar) concentrations of acrylamide used in this study, whereas its content in baked foods may approach 10(-5) M. Instead, a chronic toxicity study at its micromolar concentrations is recommended, with parallel involvement of thiol compounds. The data on the reaction rates of acrylamide with the examined thiols can be obtained. This would facilitate the characterization of the protective effects - is this a result of the direct (chemical) reactions, or the repair of proteins/DNA after their exposure to ROS or formation of covalent adducts with acrylamide. Besides, the authors should discuss the difference between their current and previous [39] data.
Specific and minor points:
Abstract: #19 - disulfide-reducing;
Introduction: #40 - commas; #65 - what is ER? #66-82 - the text is simply descriptive, may be condensed; #84 - carbohydrates;
Results: Unclear meaning of #199-202; The text frequently duplicates the legends to the figures, e.g. Figs. 2,3,5. Besides, the same phrases are used in several subsections for the description of the treatment. Figures 4F and 10E (morphology) may be shown together comprising separate figure. The data on the effects on some downstream proteins (#351-355) are not analyzed in the Discussion.
Discussion: this section is mostly descriptive (repetition of the text of Results section) with little or too well-hidden additional value.
Author Response
The authors performed a solid study on the protective effects of certain thiols against acrylamide cytotoxicity, the later expressed as the suppression of its action on a number of cell signaling proteins. However, the study possesses some shortcomings which limits its significance (detailed below).
Detail comments
The main problematic point is too high (millimolar) concentrations of acrylamide used in this study, whereas its content in baked foods may approach 10(-5) M.
Instead, a chronic toxicity study at its micromolar concentrations is recommended, with parallel involvement of thiol compounds. The data on the reaction rates of acrylamide with the examined thiols can be obtained. This would facilitate the characterization of the protective effects - is this a result of the direct (chemical) reactions, or the repair of proteins/DNA after their exposure to ROS or formation of covalent adducts with acrylamide.
Reply
We agree with the reviewer that in baked foods ACR is present at micromolar rang concentrations, and for reversing these effects one should design long-term experiments that require a chronic approach. Indeed, such studies have been performed (see reference #14)
However, the present study was aimed at reversing ACR toxicity that occurs during a large industrial spill. We addressed the possibilities for overcoming toxicity caused by large amounts of ACR, which would be relevant to overcoming acute industrial disasters, including terror attacks
Besides, the authors should discuss the difference between their current and previous [39] data.
Reply According to the reviewer request we added to the discussion the difference between our current and previous data. The data published in our earlier study described the potency of NAC, AD4 and TXM peptides in reversing oxidative stress induced by inhibiting thioredoxin reductase. The present study shows first that ACR, activates three different MAPK pathways that can be inhibited by TXM-peptides and to a lesser extent by AD4/NACA and NAC. These antioxidants appear to inhibit ACR-induced activation of the MAPK pathways, may indicate suggest that ACR generates oxidative stress through generating Michael adducts with different redox proteins including thioredoxin and GSH.
Specific and minor points:
Abstract: #19 - disulfide-reducing; was corrected
Introduction: #40 - commas; was corrected
#65 - what is ER? Endoplasmic reticulum, was inserted
#66-82 - the text is simply descriptive, may be condensed; the text was condensed, as suggested #84 - carbohydrates; was corrected
Results: Unclear meaning of #199-202;
The text frequently duplicates the legends to the figures, e.g. Figs. 2,3,5. We omitted the duplicate sentences
Besides, the same phrases are used in several subsections for the description of the treatment. Figures 4F and 10E (morphology) may be shown together comprising separate figure. The data on the effects on some downstream proteins (#351-355) are not analyzed in the Discussion.
Discussion: this section is mostly descriptive (repetition of the text of Results section) with little or too well-hidden additional value.

Reviewer 3 Report
The authors tested the ability of single cysteine, double cysteine, triple cysteine, and triple cysteine Levodopa-containing peptide antioxidants for their ability to prevent MAPK activation in response to acrylamide administration to cell lines in culture. The experimental design is good and the paper is fairly well-written. The data analysis and statistical approach analysis is appropriate. However, I have two major concerns.
Major comments:
Comment 1: The major flaw of the paper is that the authors appear to assume that MAPK activation is the main mechanism of toxicity of ACR in the PC12 cells without showing data or referencing a paper showing this. If no reference is available, the authors need to show that a MAPK inhibitor will decrease the toxicity of ACR on the PC12 cells as the first experiment in the Results section. If the authors cannot show this, their use of PC12 cells is not relevant to the mechanism of ACR toxicity.
Comment 2: The second largest flaw in the manuscript is the use of PC12 cells to model the toxicity of ACR to substantia nigra neurons in Parkinson’s disease. A dopaminergic cell line should be used in addition to the preliminary results with PC12 cells that directly compare the effects of SuperDopa to the other sulfhydryl-containing peptides.
Comment 3: Limitations of the study should be clearly stated at the end of the Discussion.
Line 98: auranofin induced -> auranofin (thioredoxin reductase inhibitor)-induced
Line 175: dependent -> dependence
Line 198: remove -> it was removed
Line 199: Cy -> Cys
Line 200: in flanked -> flanked
Line 200: residues,. -> residues. [remove comma]
Line 213: ERK1 phosphorylation IC50 of 131 ± 0.4 μM, ERK2 IC50 of 133 ± 0.4 μM, and JNK IC50 of 102 ± 22 μM -> The IC50 for ERK1 was 131 ± 0.4 μM, the IC50 of ERK2 was 133 ± 0.4 μM, and the IC50 of JNK was 102 ± 22 μM
Figure 3 panel A: Acetyl-cysteine -> N-acetylcysteine
Line 218: Cell lysates proteins in equal amounts -> Equal amounts of cell lysate protein
Line 222: **P value -> **p
Figure 4A: Acetylcysteine amide -> N-acetylcysteine amide
Line 263: 0.2mM -> 0.2 mM
Line 283: (AcC-x-x-C 282 NH2). -> (AcC-x-x-C 282 NH2) [remove period]
Line 303: ERK1and -> ERK1 and
Line 353: RSK -> ribosomal S6 kinase (RSK)
Line 405: antidots -> antidotes
Line 406: residues -> residues,
Line 407: molecule -> molecules
Line 407: antidot -> antidotes
Line 409: two fold -> two-fold
Line 417: or -> or decreased activity
Lines 418 and 420: 5mM -> 5 mM [Be consistent with a space in between number and mM throughout manuscript]
Line 426: thiol based -> thiol-based
Line 445: ACR induced -> ACR-induced
Line 446: the NAC -> NAC
Line 448: MAPKs -> MAPK phosphorylation
Line 448: membrane -> increased membrane
Line 449: opposed to the membrane impermeable -> when compared to the lower membrane permeability of
Line 450: ACR -> ACR-induced
Line 468: stress induced -> stress-induced
Line 470: Traumatic Brain Injury -> traumatic brain injury
Line 486: TXM-peptides -> TXM-peptide
Line 495: damaging -> damage
Line 498: neuron -> neurons
Line 507: photosensitized induced -> photosensitization-induced
Lines 508 and 512: SD like -> SD-like
Line 519: ACR -> ACR-induced
Line 522: peptides-namely, -> peptides, namely
Line 531: thiol dependent -> thiol-dependent
Line 533: ACR-antidotes -> ACR antidote
Line 535: PD like -> PD-like
Author Response
comments
The authors tested the ability of single cysteine, double cysteine, triple cysteine, and triple cysteine Levodopa-containing peptide antioxidants for their ability to prevent MAPK activation in response to acrylamide administration to cell lines in culture. The experimental design is good and the paper is fairly well-written. The data analysis and statistical approach analysis is appropriate. However, I have two major concerns.
Major comments:
Comment 1: The major flaw of the paper is that the authors appear to assume that MAPK activation is the main mechanism of toxicity of ACR in the PC12 cells without showing data or referencing a paper showing this. If no reference is available, the authors need to show that a MAPK inhibitor will decrease the toxicity of ACR on the PC12 cells as the first experiment in the Results section. If the authors cannot show this, their use of PC12 cells is not relevant to the mechanism of ACR toxicity.
Reply
Please note that ACR-induced neurotoxicity through the activation of the MAPK has been shown in several studies including PC12 cells. We have included references to these studies in the introduction; Please see references # 13-18
Also, in vivo, oral administering of ACR to rats through gastric gavage showed a significant effect on p38MAPK ; Please see reference “Environ Sci Pollut Res Int . 2021 Sep;28(36):49808-49819. doi: 10.1007/s11356-021-14049-4 “ /
We do not suggest that the primary mechanism of ACR toxicity occurs through MAPK inhibition. Instead, we investigated whether ACR-induced activation of MAPK pathways, known to initiate inflammatory apoptotic cell death, could be inhibited by thiol-based compounds. Our findings demonstrate that various thiol reagents, which are recognized for their ability to inhibit MAPK activity and prevent cell death, also reduce ACR-induced MAPK activation in PC12 cells
Comment 2: The second largest flaw in the manuscript is the use of PC12 cells to model the toxicity of ACR to substantia nigra neurons in Parkinson’s disease. A dopaminergic cell line should be used in addition to the preliminary results with PC12 cells that directly compare the effects of SuperDopa to the other sulfhydryl-containing peptides.
Reply
I completely agree with the reviewer’s comment that an additional neuronal-like cell line would be more appropriate for modeling Parkinson's disease. However, it is important to note that we do not use PC12 cells as a model for assessing ACR toxicity in substantia nigra neurons related to Parkinson’s disease. Rather, we mention that ACR, which has been associated with Parkinsonian symptoms, may have an additional advantage in case of acute ACR toxicity by the use of SD, which releases levodopa during proteolysis
Comment 3: Limitations of the study should be clearly stated at the end of the Discussion.
Reply
According to the reviewee suggestion we jave added a section of limitations of the study
Limitation of the study
Unlike chronic toxicity studies that focus on the micromolar concentrations of ACR typically found in food, this study examines the effects of acute exposure to higher concentrations of ACR. The scope is confined to the three MAPK pathways and does not extend to other ACR-modified cellular processes. It remains important to determine whether the thiol-based antioxidants directly interact with or scavenge ACR to prevent ROS production, or if their effects on MAPKs are due to scavenging a secondary increase in ROS
Detail comments :
Reply
We are really thankful for the excellent corrections made by the reviewer.
We have corrected all of them.
Line 98: auranofin induced -> auranofin (thioredoxin reductase inhibitor)-induced
Line 175: dependent -> dependence
Line 198: remove -> it was removed
Line 199: Cy -> Cys
Line 200: in flanked -> flanked
Line 200: residues,. -> residues. [remove comma]
Line 213: ERK1 phosphorylation IC50 of 131 ± 0.4 μM, ERK2 IC50 of 133 ± 0.4 μM, and JNK IC50 of 102 ± 22 μM -> The IC50 for ERK1 was 131 ± 0.4 μM, the IC50 of ERK2 was 133 ± 0.4 μM, and the IC50 of JNK was 102 ± 22 μM
Figure 3 panel A: Acetyl-cysteine -> N-acetylcysteine
Line 218: Cell lysates proteins in equal amounts -> Equal amounts of cell lysate protein
Line 222: **P value -> **p
Figure 4A: Acetylcysteine amide -> N-acetylcysteine amide
Line 263: 0.2mM -> 0.2 mM
Line 283: (AcC-x-x-C 282 NH2). -> (AcC-x-x-C 282 NH2) [remove period]
Line 303: ERK1and -> ERK1 and
Line 353: RSK -> ribosomal S6 kinase (RSK)
Line 405: antidots -> antidotes
Line 406: residues -> residues,
Line 407: molecule -> molecules
Line 407: antidot -> antidotes
Line 409: two fold -> two-fold
Line 417: or -> or decreased activity
Lines 418 and 420: 5mM -> 5 mM [Be consistent with a space in between number and mM throughout manuscript]
Line 426: thiol based -> thiol-based
Line 445: ACR induced -> ACR-induced
Line 446: the NAC -> NAC
Line 448: MAPKs -> MAPK phosphorylation
Line 448: membrane -> increased membrane
Line 449: opposed to the membrane impermeable -> when compared to the lower membrane permeability of
Line 450: ACR -> ACR-induced
Line 468: stress induced -> stress-induced
Line 470: Traumatic Brain Injury -> traumatic brain injury
Line 486: TXM-peptides -> TXM-peptide
Line 495: damaging -> damage
Line 498: neuron -> neurons
Line 507: photosensitized induced -> photosensitization-induced
Lines 508 and 512: SD like -> SD-like
Line 519: ACR -> ACR-induced
Line 522: peptides-namely, -> peptides, namely
Line 531: thiol dependent -> thiol-dependent
Line 533: ACR-antidotes -> ACR antidote
Line 535: PD like -> PD-like

Round 2
Reviewer 1 Report
The authors have addressed the majority of my concerns from the previous review. The cytotoxicity experiments reveal that the effect of thiol-based antioxidants on overall ACR-induced cell death is marginal, but the effects on MAPK activation are compelling. I recommend proceeding with publication.
One minor correction in the opening sentence of the abstract: it should be 'low molecular weight' rather than 'small molecular weight.'
Author Response
Response to reviewer #1
The reviewer appeared to be satisfies with our response

Reviewer 2 Report
The Discussion section was little changed and remains descriptive but not analytical. This should be corrected.
The new data (Fig. 5C and partly Fig. 5B) do not correspond to their description in subsection 3.5. This should be corrected.
Author Response
Response to reviewer#2
We corrected small to low in the first sentence of the abstract

Reviewer 3 Report
The authors made most of the important suggested changes from the original review. Only minor formatting changes are required.
Major comments: none
Minor comments: Wording and capitalization
Please be consistent with the spacing before the mM and ml abbreviations after numbers, i.e 5mM or 5 mM.
Lines 14, 25, 96, 301, 306, 399, 403, 404, 435, 450, 487, and 522: MAPKs -> MAPK
Line 51: glycinamide -> glycidamide [glycinamide is not a reactive epoxide]
Line 141: ACR 2.5 mM -> 2.5 mM ACR
Line 144: X3 -> 3X
Line 148: a630nM in -> an absorbance of 630 nm in a
Line 152: gel -> gels
Line 168: Delete the word “depndence” [or change to Time-Dependence]
Line 179: 1, -> 1 [remove the comma]
Line 185: dependent -> dependence
Line 247: ACR -induced -> ACR-induced [remove the space]
Line 257: prevent ACR induced -> Prevent ACR-Induced
Line 281: toxicity on cell viability -> Toxicity on Cell Viability
Line 281: NAc -> NAC
Line 291: incubated -> incubated with
Line 294: ACR 2.5 mM -> 2.5 mM ACR
Line 300: NH2). -> NH2)
Line 325: by the -> by
Line 439: MAPK’s -> MAPK
Line 487: di-thiol -> Di-thiol
Line 518: ROS -> ROS.
Line 520: prevent -> prevent
Line 536: disease -> disease.
Author Response
Reviewer 3
Major comments
The authors made most of the important suggested changes from the original review. Only minor formatting changes are required.
Major comments: none
Detail comments
Minor comments: Wording and capitalization
Please be consistent with the spacing before the mM and ml abbreviations after numbers, i.e 5mM or 5 mM.
We made all the spacing before mM and ml the same throughout the ms.
Lines 14, 25, 96, 301, 306, 399, 403, 404, 435, 450, 487, and 522: MAPKs -> MAPK
was changed to MAPK
Line 51: glycinamide -> glycidamide [glycinamide is not a reactive epoxide]
Was corrected
Line 141: ACR 2.5 mM -> 2.5 mM ACR
Was corrected
Line 144: X3 -> 3X
Was corrected
Line 148: a630nM in -> an absorbance of 630 nm in a
Was corrected
Line 152: gel -> gels
Was deleted
Line 168: Delete the word “depndence” [or change to Time-Dependence]
Was changed to Time-dependence
Line 179: 1, -> 1 [remove the comma] comma was removed
Line 185: dependent -> dependence was changed to dependence
Line 247: ACR -induced -> ACR-induced [remove the space] space was removed
Line 257: prevent ACR induced -> Prevent ACR-Induced was corrected
Line 281: toxicity on cell viability -> Toxicity on Cell Viability was corrected
Line 281: NAc -> NAC was corrected
Line 291: incubated -> incubated with was corrected
Line 294: ACR 2.5 mM -> 2.5 mM ACR was corrected
Line 300: NH2). -> NH2) was corrected
Line 325: by the -> by was corrected
Line 439: MAPK’s -> MAPK was corrected
Line 487: di-thiol -> Di-thiol was corrected
Line 518: ROS -> ROS. was corrected
Line 520: prevent -> prevent was corrected
Line 536: disease -> disease. was corrected

Round 3
Reviewer 2 Report
The question regarding Fig. 5 remains unanswered.
In contrast to data of Fig. 5A (significant protective effect of 2 mM NAC) the data of Fig. 5B,C show a very limited effect of other compounds. Thus, #283-287 should be corrected.
Author Response
Thank you for your comments.
We would like to clarify that the experiments assessing inhibition of ACR-induced MAPK activation and cell survival were conducted with distinct protocols and conditions. Specifically, the assessment of NAC’s ability to inhibit ACR-induced ERK1/2 activation involved applying 5mM ACR for 2 hours, washing, and subsequently adding NAC at concentrations up to 0.5 mM, with an additional 1-hour incubation. Under these conditions, NAC effectively reversed ACR's effect on ERK1/2 phosphorylation at 0.5 mM, as depicted in Fig. 3C,D. Similarly, in this setup, AD4 inhibited ERK1/2 at 0.1 mM (Fig. 4C, D), and CB3 showed a complete reversal of p38 MAPK activation at 0.2 mM (Fig. 4G).
In the survival assay, cells were incubated with 2.5 mM ACR for 2 hours, washed, treated with the tested compounds, and then incubated for an additional 48 hours. Under these conditions, unlike the MAPK assay (Fig. 3C, D), NAC displayed no effect at 0.5 mM and was only effective at 2.0 mM (Fig. 5A). Similarly, AD4 (Fig. 5B) and CB3 (Fig. 5C) showed only a slight increase in cell survival at concentrations that were effective in inhibiting ACR-induced MAPK phosphorylation (Fig. 4C, D, and Fig. 4G). These discrepancies may be attributed to differences in the experimental protocols, particularly the prolonged intracellular effect of ACR over 48 hours and the overall assay time. Consequently, establishing a direct correlation between these two assays is challenging.